# Physicians' satisfaction with clinical laboratory services at public hospitals in Ethiopia: A national survey

Hiwot Amare Hailu [1]*, Anteneh Yalew[2,3], Adinew Desale[4], Habtamu Asrat[1], Sisay Kebede[1], Daniel Dejene[1], Hiwot Abebe[1], Andargachew Gashu[1], Birhan Moges[1], Nebiyou Yemanebrhane[1], Daniel Melese[1], Birhanu T. Ayele[3], Adisu Kebede[1], Ebba Abate[1]

1 Ethiopian Public Health Institute, Addis Ababa, Ethiopia, 2 Department of Statistics, College of Natural and Computational Sciences, Addis Ababa University, Addis Ababa, Ethiopia, 3 Division of Epidemiology and Biostatistics, Department of Global Health, Faculty of Medicine and Health Sciences, Stellenbosch University, Cape Town, South Africa, 4 ILEX Biotech Ltd, CRO Ethiopia, Addis Ababa, Ethiopia

* hiwotamare20@gmail.com

**Data Availability Statement:** All relevant data are within the paper and its Supporting Information files.

**Funding:** ADL is employed by and receives salary from IBEX Biotech Ltd. The funder had no role in study design, data collection and analysis, decision to publish, or preparation of the manuscript.

**Competing interests:** ADL is employed by and receives salary from IBEX Biotech Ltd. This does not alter our adherence to PLOS ONE policies on sharing data and materials.

## Abstract

### Introduction

Physician is a central figure in the client list of clinical laboratory. Monitoring physicians' satisfaction with laboratory service is an important indicator of the quality management system and required by international laboratory standards. However, there is no national data on physician satisfaction with laboratory services in Ethiopia. Therefore, the aim of this national survey was to assess satisfaction level of physicians with laboratory services at public hospitals in Ethiopia.

### Methods

Institutional based cross-sectional study design was employed from November 1-30/2017. A total of 327 physicians were randomly selected from 60 public hospitals from all regions of Ethiopia. Data was collected using pre-tested self-administered questionnaire and analyzed with SPSS version 23 software. Logistic regression model was fitted to identify predictors of physician satisfaction with laboratory services. A p-value of less than 0.05 was taken as statistically significant.

### Results

Overall, 55% of physicians were satisfied with the clinical laboratory services. More than half of the physicians were satisfied with the existing laboratory request form (69%), legibility and completeness of laboratory report (61%), notification of new test (78%) and test interruption (70%). On the other hand, many physicians were dissatisfied with the absence of laboratory hand book (87.5%), the existing test menu (68%), lab-physician interface (62%), availability of referral and/or back up service (62%), notification of Turn Around Time (TAT) (54%), timely notification of panic result (55%), long TAT (33.1%), provision of urgent service (67%), and timely advisory service (57%). Most of the physicians perceived that

consistent quality of service was not delivered at all working shifts (71%). At 5% level of significance, we did not find enough evidence to conclude that sex, age, marital status, education level, and experience were statistically associated with physician satisfaction (p-values > 0.05).

## Conclusion

This national survey revealed nearly half of the physicians were unsatisfied with laboratory service at public hospitals in Ethiopia, which mainly related to lack of adequate test menu, laboratory hand book, on time notification of panic result, provision of urgent service, timely advisory service, delivery of quality services in all working shifts and weak lab-physician interface. Therefore, hospital management should address the gaps and improve the needs of physicians for better patient health care. In addition, laboratories should evaluate and monitor physician satisfaction level at regular interval.

## Introduction

Medical laboratories are essential component and one of the most important departments at any healthcare services where medical tests and investigations are done in order to generate reliable and accurate information regarding patient's health [1]. Laboratory reports are usually the bases of medical decisions and possible management plans considered by physicians [2]. Medical laboratories have customers whose need should be addressed efficiently. Physicians, the principal client of medical laboratories initially request the services. Health care providers are expected to have access to accurate, clinically relevant information that can be understood and used in a timely manner [3].

Assuring a wide range of quality laboratory services is a challenging processes that need support from clients, clinical service providers, managers, laboratory professionals and other stakeholders. Several features of the laboratory services could be inspected from the perspective of physicians including, quality/reliability of test results, staff courtesy, accessibility of pathologist, accessibility of laboratory manager, phlebotomy services, test menu adequacy, accessibility of laboratory staff, courier services, routine test turnaround time (TAT), laboratory management responsiveness, inpatient stat test TAT, critical value notification, clinical report format, outpatient stat test TAT, and esoteric TAT [4]. Quality of laboratory results and adequate test menu remain the most important element for most physicians [5]. Physicians also need assurance of laboratory responsibility on test menu, accurate collection manual, requisition forms and assurance of working with competent personnel, validated method and good process control [3].

Previous studies showed that physicians' request behavior and treatment interventions are influenced by the communication and interactions between laboratory and clinical health workers. Lack of communication is a barrier to effective healthcare service. Improved communication between laboratory and clinical health workers could have a positive attitude to request and use laboratory diagnostic services and, eventually, quality of patient care [6].

In clinical laboratory, monitoring customers' satisfaction is an important indicator of the quality management system and required by laboratory quality standards, such as ISO 15189: 2012. Satisfaction is a judgment given by people that reflect their experience under specific circumstances, not a pre-existing phenomenon waiting to be measured [7]. It is a perception and

an attitude that a customer can have or view towards a total experience of health care services [8]. Physicians' opinions are essential components in providing laboratory managers with opportunities to identify areas for improvement [4]. Various studies investigated satisfaction of the primary healthcare providers (physicians) of laboratory services to identify possible limitations for future development [4, 8, 9].

Clinical laboratories are expected to assess physicians' satisfaction with clinical laboratory services to improve the service. However, there has no national level information or data related to physicians' satisfaction in Ethiopia. Therefore, this study aimed to assess physicians' satisfaction with clinical laboratory services in public hospitals of Ethiopia. The findings might be useful to design and implement measures to improve the quality of clinical laboratory services.

## Materials and methods

### Study design and area

The institutional based cross-sectional study design was conducted from November 1 to 30, 2017. Based on the 2017 prediction report, Ethiopia has a total population of 94,351,001 and about 80% of the population lives in rural areas. According to the 2016 Ethiopian Minister of Health report, there were 189 government hospitals with functional laboratory service, 3547 public health centers and 16447 health posts in Ethiopia. The physician to population ratio was 1:17160 [10]. These health facilities provide different clinical and laboratory services to the community. Each hospital laboratory provides different services that include ART monitoring, microbiology, parasitology, serology, electrolyte, hormone analysis, and others tests.

### Study population

All physicians, who were on duty during the data collection period, were the study population.

### Sample size and sampling procedure

The required sample size of physicians was determined by the following formula:

$$n = \text{deft}^2 \frac{(1/p - 1)}{\alpha^2}$$

Where, p is the assumed value of the population proportion of the underlying variable defining the main indicator of the survey coverage. The proportion of physician's satisfaction with laboratory services was 50% according to a study done at selected hospitals in eastern Ethiopia [11], deff is the design effect. Design effect of 2 was used in this survey, $\alpha$ is the specified relative standard error equals to 0.08 physicians, at 95% confidence level and it's a good relative precision of the indicator at domain estimate level [12]), and response rate is the expected response rate of the survey was 90% for customer survey and as individual response rate.

Accordingly, the required sample size was 348 physicians from 60 hospitals. Allocation of the total sample sizes to the regions and hospital types was considered. Since some regions and hospital types are few in size, we applied a power allocation to guarantee a sufficient sample size in small regions and hospital types in size.

### Data collection procedures

Data was collected using a pre-tested, structured and a self-administered questionnaire. The questionnaire was pre-tested in similar settings which were not included in the study. The

questionnaire contained socio-demographic characteristics, courtesy of the laboratory staff, test availability, critical value notification, courier service, the reliability of test results, provision of timely test results and others variables.

### Data quality assurance

Data collectors and supervisors were trained on how to select study participants and collect data. In-order to identify and solve the confusing points, we had pre-tested the questionnaire prior to the actual survey with pilot sites. The number of participants in pre-test was 33 (10%). They were recruited from four towns, one public hospital from each town. Regular supervision, spot checking and reviewing the completed questionnaire was carried out daily by regional supervisors. Double entry of 15% of the data was carried out.

### Data entry and analysis

Data were entered using Epi Info version 7.2 and analyzed using SPSS version 23. Descriptive statistics were computed to describe data. A 5-point Likert scale rating of very dissatisfied (1-point), dissatisfied (2-points), neutral (3-points), satisfied (4-points) and very satisfied (5-point) was used. The mean score of satisfaction for each participant was calculated as the average of all satisfaction items. A mean score of 3 and less than 3 was taken as an indicator of participants' perceived dissatisfaction and a score of more than 3 was taken as the participant was satisfied.

Binary logistic regression model was fitted to identify predictors of physicians' satisfaction with laboratory services. Those variables significant at a p-value of 0.20 in the univariate analysis were included in multiple regression model. A p-value of less than 0.05 was used to determine statistical significance. Adjusted Odds Ratio (AOR) with 95% confidence interval (CI) was used to identify factors affecting physicians' satisfaction level of laboratory customers.

### Ethical consideration

Ethical clearance was obtained from the Scientific and Ethical Review Committee (SERC) of the Ethiopian Public Health Institution (EPHI). An official permission letter was delivered to the respective regional health bureaus by EPHI during the field work. The facility administration was informed about the general objective and significance of the study through an official letter. Data were collected anonymously. For the purpose of data collection, the aim of the study was explained, and written informed consent was obtained from study participants before administering the questions. All participants were informed of their right to refuse the participation at any time.

## Results

### Socio-demographic characteristics of physicians

Three hundred forty-eight survey questionnaires were distributed, and 327 were collected. This makes the response rate was 94%. These physicians selected from 60 public hospitals in Ethiopia and 78.9% of them were male, and 42.5% were married. The median age and interquartile range of the participants were 29 and (27–32) years, respectively. Nearly, 68% of the participants had less than five years' experience and 10% of them were specialist in a different discipline (Table 1).

### Overall satisfaction level of physicians

Overall, 55% of the participated physicians were satisfied with the services provided by public hospital laboratories. More than half of the participants were satisfied with existing laboratory

**Table 1. Distribution of socio-demographic characteristics of respondents at public hospitals in Ethiopia, November 2017.**

| Characteristics | Number (n = 327) | Percent |
|---|---|---|
| **Sex** | | |
| Male | 258 | 78.9 |
| Female | 69 | 21.1 |
| **Age Group** | | |
| 24–29 | 183 | 56.0 |
| 30–40 | 130 | 39.8 |
| >40 | 14 | 4.3 |
| **Marital Status** | | |
| Single | 188 | 57.5 |
| Married | 139 | 42.5 |
| **Educational Status** | | |
| MD | 292 | 89.3 |
| Specialized | 35 | 10.7 |
| **Experience (years)** | | |
| 1–4 | 223 | 68.2 |
| ≥5 | 104 | 31.8 |

MD = Medical Doctor

request form, legibility and completeness of laboratory report, notification of new test and test interruption. On the other hand, they were dissatisfied with the availability of lab handbook, availability of test menu, availability of referral or back up service, notification of TAT, notification of panic result, provision of urgent service with a timely fashion, timely advisory/expert service and quality of service in all working shifts.

**Physicians' satisfaction with laboratory services.** As depicted in Table 3, 86.2% of the study participants were satisfied with the presence of laboratory staff at the workstation during the working hour. The finding showed 87.5% of the physicians did not receive the laboratory handbook from the laboratory. Regarding the laboratory request form, 69.42% of the physicians were comfortable and satisfied with the current request form. Most of the respondents (67.89%) were dissatisfied with the available laboratory tests to manage their patients. Nearly thirty-eight percent of the physicians had the opportunity for laboratory services with referral and/or back up laboratories. Out of these participants, 59.5% of them were comfortable with the backup services (see Tables 2 & 3).

## Physicians and laboratory communication

Regarding the lab-physician interface, nearly 38% of the physicians were satisfied with the interaction they have with laboratory personnel. Physicians were also satisfied with on time notification of a newly interoduced test (77.7%), test interruption (70%) and panic results (44.6%) by the laboratory. The survey result also indicated that 33% of the participants were dissatisfied with the provision of urgent services in a timely fashion. In addition, 43% of the participants were satisfied with the timely laboratory expert advisory service, and 42.5% of them had a positive perception for the laboratory's ability to resolve their complaints (see Tables 2 and 3).

**Physicians' satisfaction with laboratory report.** Forty-six percent of the participants had posted turnaround time of each laboratory tests, out of them, 67% received the laboratory

**Table 2. Participants' frequency and percentage distribution of laboratory services at selected public hospitals in Ethiopia, November 2017.**

| Characteristics | Number (n = 327) | Percent |
|---|---|---|
| Availability of updated Laboratory handbook | | |
| No | 286 | 87.5 |
| Yes | 41 | 12.5 |
| Presence of lab personnel at bench work | | |
| No | 45 | 13.8 |
| Yes | 282 | 86.2 |
| Availability of backup/referral Service (222) | | |
| No | 138 | 62.16 |
| Yes | 84 | 37.84 |
| Comfortable with backup service (84) | | |
| No | 34 | 40.50 |
| Yes | 50 | 59.50 |
| Availability of TAT of available tests in your work area | | |
| No | 176 | 53.8 |
| Yes | 151 | 46.2 |
| Receive laboratory report within agreed TAT (n = 151) | | |
| No | 50 | 33.1 |
| Yes | 101 | 66.9 |
| Immediate notification of panic results | | |
| No | 181 | 55.4 |
| Yes | 146 | 44.6 |
| Notification during new tests are introduced | | |
| No | 73 | 22.3 |
| Yes | 254 | 77.7 |
| On time notification during test interruption | | |
| No | 98 | 30 |
| Yes | 229 | 70 |

report with in the established turnaround time. In addition, 61% of the participants were satisfied with the legibility and completeness of laboratory reports. Seventy-one percent (232) of the participated physicians perceived that laboratory services did not have the same quality in all working shifts (day, night, holiday and weekend). Out of them, 12.5% were doubtful of the quality of the laboratory services at any time whereas 87.5% of them did not trust the quality of the laboratory services done during the over-time (night, holiday, weekend).

**Table 3. Satisfaction level of physicians with different components of laboratory services at the public hospital in Ethiopia, November 2017.**

| Characteristics | Dissatisfied number (%) | Satisfied number (%) |
|---|---|---|
| Timely expert/advisory service of the lab staff | 186 (56.88) | 141(43.12) |
| Laboratory's ability to resolve complaints | 188(57.49) | 139(42.51) |
| Laboratory's request form | 100(30.6) | 227(69.4) |
| Existing test menu | 222(67.89) | 105(32.11) |
| Legibility and completeness of laboratory report | 127 (38.84) | 200(61.16) |
| Provision of urgent services in a timely fashion | 219(67.0) | 108(33.0) |
| Lab-clinical interface | 204(62.39) | 123(37.61) |
| Satisfaction with the assistance of the handbook | 12(29.26) | 29 (70.74) |

**Table 4. Association of independent variables with a satisfaction level of physicians at selected public hospitals in Ethiopia, November 2017.**

| Characteristics | Physician satisfaction | | COR (95%CI) | AOR (95%CI) | P-value |
|---|---|---|---|---|---|
| | Dissatisfied | Satisfied | | | |
| **Sex** | | | | | |
| Male | 122 | 136 | 1.6(0.9, 2.7) | 1.6(0.91, 2.81) | 0.10 |
| Female | 25 | 44 | 1 | 1 | |
| **Age group** | | | | | |
| 24–29 | 78 | 105 | 0.7(0.2, 2.3) | | 0.39 |
| 30–40 | 64 | 66 | 0.6(0.2. 1.8) | | |
| >40 | 5 | 9 | 1 | | |
| **Marital status** | | | | | |
| Single | 81 | 107 | 1 | 1 | |
| Married | 66 | 73 | 1.19(0.77, 1.86) | 1.29(0.77, 2.16) | 0.32 |
| **Edu. status** | | | | | |
| MD | 127 | 165 | 1 | 1 | |
| Specialized | 20 | 15 | 1.73(0.85, 3.52) | 1.84(0.82, 4.15) | 0.14 |
| **Experience (yr)** | | | | | |
| 1–4 | 100 | 123 | 1 | 1 | |
| ≥5 | 47 | 57 | 1.01 (0.63, 1.62) | 1.01(0.51, 2.01) | 0.97 |

### Factors affecting satisfaction of physicians

Bivariate logistic regression was used to identify possible explanatory variables, those variables with a p-value of less than 0.20, were taken to multiple binary logistic regression model. As a result, sex (P = 0.10), age (P = 0.22), marital status (P = 0.32), educational status (P = 0.14) and experience (P = 0.97) were not significantly associated with physician overall satisfaction level with laboratory services (see Table 4).

## Discussion

Physicians are primary customers of hospital laboratory and their perception of the provided services is critical for improvement and quality service. Satisfaction survey is one of the means for them to express concerns about the services received, and to express their views about the services that need improvement. Hence, the present national survey tries to assess the physicians' satisfaction with laboratory services at public hospitals in Ethiopia.

In this national survey, 55% of the physicians were satisfied with the services provided by public hospital laboratories. This overall satisfaction rate is nearly similar with reports from St. Paulo's Hospital Millennium Medical College (60%), University of Gondar Hospital (51.5%), Pusan National University Hospital (58.1%), and Saudi Arabia (53.3%) [13, 14, 15, 2]. It is lower than findings from Nekemte Referral Hospital (65%), and selected hospitals in eastern part of Ethiopia (80%), and College of American Pathologists Q-Probes study of 81 Institutions in 2016 [16, 11, 5]. The difference may be due to difference in participants and sample size. In our study, all study participants were physicians however in the other studies, participants were health care providers (nurses, health officers, physicians).

The interaction between physicians and laboratory personnel is mandatory for better patient health care. They may communicate face-to-face or by request and report, memos, standard operating procedures, manuals, phone calls, text messages, e-mails or computerized system. In this study, most of physicians were satisfied with legibility and completeness of laboratory test report (61.16%), notification of new test introduction (77.70%), notification of test

interruption due to different reasons (70%) and availability of standard laboratory request form (69.42%). Previous studies have identified lack of communication as a barrier to effective healthcare [6, 17, 18, 19, 20]. Improved communication between clinicians and laboratory workers is essential to changing clinicians' attitudes about the reliability of diagnostic tests, possibly leading to increased use of laboratory diagnostics and, ultimately, improving patient care [6].

Physicians need a wide range of test menu in the laboratory to manage their patients. In the current study, nearly 67.89% of the physicians were dissatisfied with the existing test menu in the laboratory. This finding showed services provided by public hospital laboratories were not adequate to fulfill expectations of the physicians. This finding was supported by reports from Tanzanya, Egypt and Korea. Previous studies conducted in Gondar University hospital, and Nekemte referral hospital in Ethiopia showed that physicians perceived that the existing test menu was not adequate [21, 22, 15, 14, 16].

Physicians need updated laboratory handbook that is complete and user-friendly. This study showed 87.5% of physicians did not have laboratory handbook. It is an ISO requirment that the laboratory should have information available for users and patients of the laboratory services though laboratory hand book. It provides information about the location of the laboratory, list of tests, working hours of the laboratory, sample collection and handling requirements, biological reference intervals, instructions for completion of the request form, the laboratory's criteria for accepting and rejecting samples etc [23].

Additionally, the finding showed nearly 30.6% of the physicians were dissatisfied with the current available laboratory request form. The request form is the major means of communication between the health care provider and the laboratory. Therefore, the design of the request form must have sufficient space to support this communication. It has been allowed to provide the necessary information that include patient identification, name or other unique identifier of clinician who request examination, clinically relevant information about the patient and the request, for examination performance and result interpretation purposes, date and, where relevant, time of primary sample collection, and date and time of sample receipt [23].

Critical value intervals have been established in accordance with published information and physician are notified immediately when examination results fall within established alert or critical intervals. In this study, 55.4% of the physicians were not notified the panic result timely, and 30.89% of physicians were not provided the urgent services with a timely fashion. In addition, nearly 54% of the respondents were not aware about turnaround time of each tests performed by the laboratories. Turnaround time is one of the most noticeable signs of laboratory service and is used by many physicians to judge the quality of the laboratory. Previous studies from Tanzania, Alexanderia, Korea, Gondar University Hospital, Nekemte referral hospital and selected hospitals in east Ethiopia region, and College of American Pathologists reported similar finding that physicians were not satisfied with timely notification of panic results and provision of urgent services in a timely fashion [21, 22, 15, 14, 16, 18, 5]. The issue of ineffective communication between physicians and laboratory staff on patient care remains unresolved, therefore, there is a need of active communication between laboratory and physicians in any case of patient care activities.

The current study also showed that 71% of physicians' perceived quality of laboratory service was inconsistent in all working shifts. This finding was consistent with reports from Tanzania, Alexanderia, Gondar University Hospital, Nekemte referral hospital, and selected hospitals in east Ethiopia region [21, 22, 14, 16, 18].

Overall, none of the socio-demographic characteristics of the physicians had statistically significant association with overall satisfaction. This findings go in line with a study from

Egypt where the level of physician satisfaction was unrelated to age, gender, specialty, and work experience [24].

In Ethiopia, medical laboratories in public hospitals are directed by laboratory professionals. Laboratory personnel get the required technical and managerial trainings and assessed their competency regularly based ISO 15189:2012 requirements.

## Limitation

This study did not use open ended questionnaire to grasp additional information about laboratory services. In addition, it did not included the satisfaction of physicians working in the private health facilities. It also lacks other laboratory customers' satisfaction level.

## Conclusion

This nationwide survey report showed that nearly half of physicians were not satisfied with the services provided by public hospital laboratories. Lack of laboratory handbook, inadequate test menu, lack of on time notification of panic result and provision of urgent service with a timely fashion, lack of timely advisory/expert service and inconsistent quality of service in all working shifts contributed to the observed low satisfaction rate. Hospital laboratory actions should meet the needs of physicians that include the identified gaps. The communication between physicians and laboratory personnel should be strengthened as well as the laboratory should conduct regular satisfaction survey to identify, improve and provide feedback for continuous quality service improvement. In addition, other responsible bodies in each level should act on the identified gaps and improve the need of physicians in each hospital laboratory.

This national survey is first of its kind in Ethiopia and provided credible evidence that might be used to improve the quality of laboratory service and enhancing physicians' satisfaction. Finding of this study might serve as a baseline data for any intervention designed to improve the quality of laboratory service in the country.

## Supporting information

**S1 Data.**
(DOCX)

## Acknowledgments

The authors would like to acknowledge EPHI management for their follow up and support, all regional laboratories, EPHI quality improvement and accreditation team, all survey participants, the field staff who involved in data collection and supervision for the crucial roles played in achieving the survey goal.

## Author Contributions

**Conceptualization:** Hiwot Amare Hailu.

**Data curation:** Habtamu Asrat, Sisay Kebede, Daniel Dejene, Hiwot Abebe, Andargachew Gashu, Birhan Moges, Nebiyou Yemanebrhane, Daniel Melese.

**Formal analysis:** Hiwot Amare Hailu, Anteneh Yalew, Adinew Desale, Habtamu Asrat, Sisay Kebede, Daniel Dejene, Hiwot Abebe, Andargachew Gashu, Birhan Moges, Nebiyou Yemanebrhane, Daniel Melese, Birhanu T. Ayele, Adisu Kebede, Ebba Abate.

**Methodology:** Hiwot Amare Hailu, Anteneh Yalew, Adinew Desale, Birhanu T. Ayele.

**Project administration:** Hiwot Amare Hailu, Adisu Kebede, Ebba Abate.

**Resources:** Adisu Kebede, Ebba Abate.

**Software:** Hiwot Amare Hailu, Anteneh Yalew, Birhanu T. Ayele.

**Supervision:** Hiwot Amare Hailu, Anteneh Yalew, Adinew Desale, Habtamu Asrat, Sisay Kebede, Daniel Dejene, Hiwot Abebe, Andargachew Gashu, Birhan Moges, Nebiyou Yemanebrhane, Daniel Melese.

**Validation:** Hiwot Amare Hailu, Anteneh Yalew.

**Visualization:** Adisu Kebede, Ebba Abate.

**Writing – original draft:** Hiwot Amare Hailu, Anteneh Yalew, Adinew Desale.

**Writing – review & editing:** Anteneh Yalew, Adinew Desale, Habtamu Asrat, Sisay Kebede, Daniel Dejene, Hiwot Abebe, Andargachew Gashu, Birhan Moges, Nebiyou Yemanebrhane, Daniel Melese, Birhanu T. Ayele, Adisu Kebede, Ebba Abate.

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
