## [Decision Letter · Decision Letter 0]

9 Jan 2020

PONE-D-19-27181

Physicians’ Satisfaction with Clinical Laboratory Services at Public Hospitals in Ethiopia: A National Survey

PLOS ONE

Dear Dr Hiwot Amare

Thank you for submitting your manuscript to PLOS ONE. After careful consideration, we feel that it has merit but does not fully meet PLOS ONE’s publication criteria as it currently stands. Therefore, we invite you to submit a revised version of the manuscript that addresses the points raised during the review process.

We would appreciate receiving your revised manuscript by 31/01/2020. To enhance the reproducibility of your results, we recommend that if applicable you deposit your laboratory protocols in protocols.io, where a protocol can be assigned its own identifier (DOI) such that it can be cited independently in the future. For instructions see: http://journals.plos.org/plosone/s/submission-guidelines#loc-laboratory-protocols

We look forward to receiving your revised manuscript.

Kind regards,

Massimo Ciccozzi

Academic Editor

PLOS ONE

Journal Requirements:

2. Thank you for stating the following in the Competing Interests:

The authors have declared that no competing interests exist.

We note that one or more of the authors have an affiliation to the commercial funders of this research study : IBEX Biotech Ltd.

3. Please include additional information regarding the survey or questionnaire used in the study and ensure that you have provided sufficient details that others could replicate the analyses. For instance, if you developed a questionnaire as part of this study and it is not under a copyright more restrictive than CC-BY, please include a copy, in both the original language and English, as Supporting Information. Additionally, please provide details about the pretesting of this questionnaire - i.e. how many participants were involved and from where were they recruited?

4. Please ensure you have thoroughly discussed any limitation of this study within the Discussion section.

5. We noticed you have some minor occurrence of overlapping text with the following previous publication(s), which needs to be addressed:

https://www.scribd.com/document/377881161/lqms-en-pdf

http://www.ajrbps.com/article/PHYSICIANS%E2%80%99%20SATISFACTION%20WITH%20LABORATORY%20SERVICES%20AT%20KING%20FAISAL%20HOSPITAL%20IN%20MAKKAH,%20SAUDI%20ARABIA.pdf

https://ajlmonline.org/index.php/ajlm/article/view/126

The text that needs to be addressed involves large parts of the Introduction.

In your revision ensure you cite all your sources (including your own works), and quote or rephrase any duplicated text outside the methods section. Further consideration is dependent on these concerns being addressed.

6. Please amend either the title on the online submission form (via Edit Submission) or the title in the manuscript so that they are identical.

Reviewers' comments:

Reviewer's Responses to Questions

**Comments to the Author**

1. Is the manuscript technically sound, and do the data support the conclusions?

Reviewer #1: Yes

Reviewer #2: Yes

2. Has the statistical analysis been performed appropriately and rigorously? 

Reviewer #1: Yes

Reviewer #2: Yes

3. Have the authors made all data underlying the findings in their manuscript fully available?

Reviewer #1: Yes

Reviewer #2: Yes

4. Is the manuscript presented in an intelligible fashion and written in standard English?

Reviewer #1: No

Reviewer #2: Yes

5. Review Comments to the Author

Reviewer #1: This is an important study that has the potential to lead to significant, directed improvements in Laboratory Services in Ethiopia.

Some revisions would strengthen the paper and are necessary for a better understanding of the methodology and results.

1. The authors present data for n=327 physicians but they do not indicate the total number of surveys that were distributed and the overall response rate. The study population is listed as 'all physicians who were on duty at the data collection period" but a specific number should be included. How many surveys were distributed? What was the response rate? Was 100% participation mandated by the 'regional supervisors?' Additional information is needed to clarify this. Previous studies suggest that response rates to surveys can be quite low (30%). In their formula, the authors included a response rate of 90% but this was presented as a constant included in the equation - it is unclear if this represents the actual experienced response rate. This entire section should be clarified. In addition, the explanation of the equation for required sample size needs revision. deft^2=deff=2 is unclear.

2. References 4 and 8 are the same paper.

3. The authors have omitted or chosen not to include Larry Massie's group study of physician satisfaction surveys from 2016 in their reference list. This is the follow up paper to the paper the authors list as reference 4/8.

4. The inclusion of marital status for the physicians seems unlikely to be related to their perception of laboratory quality. It is unclear why this demographic was included.

5. The authors should include in the discussion information about how laboratories are directed in Ethiopia. Are they run by physicians or laboratory professionals? What specific training is required for staff and leadership? Could the current study be compared to other results of physician satisfaction based on the training level of the staff and leadership in the laboratory?

6. There are many grammatical errors in this manuscript that will require revision prior to publication. For example, there are five grammatical errors in the abstract's "Introduction" three-sentence paragraph.

Reviewer #2: Dear Authors,

even though the article is well done from a statistical point of view, the aim and the results of the article is more focused on clinician's satisfaction of the hospital rather than to give useful information for the international scientific community. I am not questioning the quality of the article (that is well-written) but the topic that is not internationally relevant and is more focused of clinician satisfaction than on the patients' care quality, so I suggest you to choose a different journal. If you don't agree with the review and there is something that you think I have not considered please fell free to let me know so we can discuss it together.

thank you so much for the opportunity to read your work

6. PLOS authors have the option to publish the peer review history of their article (what does this mean?). If published, this will include your full peer review and any attached files.

Reviewer #1: No

Reviewer #2: No

---

## [Author Response · Author response to Decision Letter 0]

1 Feb 2020

Response to Reviewers:

Title: Physicians’ satisfaction with clinical Laboratory Services at Public Hospitals in Ethiopia: A National Survey

Dear Prof Massimo Ciccozzi, Academic Editor;

Thank you for your giving us the chance to address comments from our reviewers. We appreciate the constructive comments from our two reviewers. We addressed the key concerns and tried to provide point-by-point replies.

For Editor Comments

Journal Requirements:

• We have revised the manuscript based on the PLOS ONE's style requirements and amend the title, authors, and affiliations in the first page as well as level 1, 2, and 3 headings of the main body.

2. Thank you for stating the following in the Competing Interests:

The authors have declared that no competing interests exist.

We note that one or more of the authors have an affiliation to the commercial funders of this research study: IBEX Biotech Ltd.

• Thanks for pointing this out. Yes, Mr. Adino was working for the Ethiopian Public Health Institute at the time the survey was conducted. He left after the data collection was finalized. Currently, he is working for IBEX Biotech Ltd. There is no any commercial funders for this study.

• The funder of this study, Ethiopian Public Health Institute, provided support in the form of salaries for authors [HAH, AY, AD, HA, SK, HA, AG, BM, NY, DM, BA, AK and EA], but did not have any additional role in the study design, data collection and analysis, decision to publish, or preparation of the manuscript. The specific roles of these authors are articulated in the ‘author contributions’ section.

• There is no any update.

• We have developed the questionnaire for this survey so we include the questionnaire as supporting information. 

Additionally, please provide details about the pretesting of this questionnaire - i.e. how many participants were involved and from where were they recruited?

• The following clarification is included in the revised manuscript. “In-order to identify and solve the confusing points, we had pre-tested the questionnaire prior to the actual survey with pilot sites. The number of participants in pre-test was 33 (10%). They were recruited from four towns, one public hospital from each town.”

• It is addressed under the topic of Methods/data collection and data quality assurance. 

4. Please ensure you have thoroughly discussed any limitation of this study within the Discussion section.

• Dear editors, we have included the limitation of this study under the discussion section of the revised manuscript.

5. We noticed you have some minor occurrence of overlapping text with the following previous publication(s), which needs to be addressed:

https://www.scribd.com/document/377881161/lqms-en-pdf

http://www.ajrbps.com/article/PHYSICIANS%E2%80%99%20SATISFACTION%20WITH%20LABORATORY%20SERVICES%20AT%20KING%20FAISAL%20HOSPITAL%20IN%20MAKKAH,%20SAUDI%20ARABIA.pdf

https://ajlmonline.org/index.php/ajlm/article/view/126

The text that needs to be addressed involves large parts of the Introduction.

In your revision ensure you cite all your sources (including your own works), and quote or rephrase any duplicated text outside the methods section. Further consideration is dependent on these concerns being addressed.

• We have revised and corrected the overlapping texts and cited all of our resources in the revised version of the manuscript. 

6. Please amend either the title on the online submission form (via Edit Submission) or the title in the manuscript so that they are identical.

• We have amend the title on the manuscript as “Physicians’ satisfaction with clinical Laboratory services at Public Hospitals in Ethiopia: A National Survey”

For Reviewers' comments:

Reviewer's Responses to Questions

Comments to the Author

1. Is the manuscript technically sound, and do the data support the conclusions?

Reviewer #1: Yes

Reviewer #2: Yes

2. Has the statistical analysis been performed appropriately and rigorously?

Reviewer #1: Yes

Reviewer #2: Yes

3. Have the authors made all data underlying the findings in their manuscript fully available?

Reviewer #1: Yes

Reviewer #2: Yes

4. Is the manuscript presented in an intelligible fashion and written in standard English?

Reviewer #1: No

Reviewer #2: Yes

5. Review Comments to the Author

Reviewer #1: This is an important study that has the potential to lead to significant, directed improvements in Laboratory Services in Ethiopia.

Some revisions would strengthen the paper and are necessary for a better understanding of the methodology and results.

1. The authors present data for n=327 physicians but they do not indicate the total number of surveys that were distributed and the overall response rate. The study population is listed as 'all physicians who were on duty at the data collection period" but a specific number should be included. How many surveys were distributed? What was the response rate? Was 100% participation mandated by the 'regional supervisors?' Additional information is needed to clarify this. Previous studies suggest that response rates to surveys can be quite low (30%). In their formula, the authors included a response rate of 90% but this was presented as a constant included in the equation - it is unclear if this represents the actual experienced response rate. This entire section should be clarified. In addition, the explanation of the equation for required sample size needs revision. deft^2=deff=2 is unclear.

• Thank you for the valuable comment. The estimated total number of physicians in public hospitals was 4204. Three hundred forty-eight survey questionnaires were distributed, and 327 were collected. This makes the response rate was 94%. Response rate for surveys can be quite low but previous studies conducted in Ethiopia among Physicians reported response rates as high as 90%. 

• Thanks for pointing this out. deff represent the design effect, which is an adjustment for the stratified sampling method used in this study. 

2. References 4 and 8 are the same paper.

• Dear reviewer, thank you for pointing this out. You are right the two references are the same. We updated this in the revised manuscript. 

3. The authors have omitted or chosen not to include Larry Massie's group study of physician satisfaction surveys from 2016 in their reference list. This is the follow up paper to the paper the authors list as reference 4/8.

• Thanks for the suggestion. We included the Larry Massie's group study of 2016 as reference 5 in the new revised manuscript. 

4. The inclusion of marital status for the physicians seems unlikely to be related to their perception of laboratory quality. It is unclear why this demographic was included.

• We fully agree. None of the socio-demographic characteristics including marital status are not associated with physicians’ satisfaction level. However, we presented all the findings as a resource to our readers. 

5. The authors should include in the discussion information about how laboratories are directed in Ethiopia. Are they run by physicians or laboratory professionals? What specific training is required for staff and leadership? Could the current study be compared to other results of physician satisfaction based on the training level of the staff and leadership in the laboratory?

• Dear reviewer, we included clarification on how laboratories are directed in Ethiopia in the last paragraph of discussion section of the revised manuscript. 

In Ethiopia, medical laboratories in public hospitals are directed by laboratory professionals not physicians. Quality management system, leadership, assigned work processes and procedures, applicable laboratory information system, health and safety, including the prevention or containment of the effects of adverse incidents, ethics, and confidentiality of patient information are provided to laboratory professionals.

The current study did not evaluate physicians’ satisfaction with training and leadership status of laboratory professionals.

6. There are many grammatical errors in this manuscript that will require revision prior to publication. For example, there are five grammatical errors in the abstract's "Introduction" three-sentence paragraph.

• Dear reviewer, thank you very much for your contractive comment. We thoroughly revise the manuscript for grammatical errors.

Reviewer #2: Dear Authors,

even though the article is well done from a statistical point of view, the aim and the results of the article is more focused on clinician's satisfaction of the hospital rather than to give useful information for the international scientific community. I am not questioning the quality of the article (that is well-written) but the topic that is not internationally relevant and is more focused of clinician satisfaction than on the patients' care quality, so I suggest you to choose a different journal. If you don't agree with the review and there is something that you think I have not considered please fell free to let me know so we can discuss it together.

thank you so much for the opportunity to read your work

• Dear reviewer, thanks for reading our manuscript and your comments.

• We still believe that this could be of an interest to the international scientific community as it provides useful information on clinician’s satisfaction in resource-limited settings. Clinician’s satisfaction plays a significant role in patient’s care quality and improving the health care. Understanding clinician’s level of satisfaction and factors affecting it might help to improve health care. Hope this convince our reviewer to reconsider his/her decision. 

6. PLOS authors have the option to publish the peer review history of their article (what does this mean?). If published, this will include your full peer review and any attached files.

Do you want your identity to be public for this peer review? For information about this choice, including consent withdrawal, please see our Privacy Policy.

Reviewer #1: No

Reviewer #2: No

---

## [Editor Report · Decision Letter 1]

9 Apr 2020

Physicians’ Satisfaction with Clinical Laboratory Services at Public Hospitals in Ethiopia: A National Survey

PONE-D-19-27181R1

Dear Dr. Hiwot Amare Hailu,

We are pleased to inform you that your manuscript has been judged scientifically suitable for publication and will be formally accepted for publication once it complies with all outstanding technical requirements.

With kind regards,

Massimo Ciccozzi

Academic Editor

PLOS ONE
---

## [Editor Report · Acceptance letter]

20 Apr 2020

PONE-D-19-27181R1 

Physicians’ Satisfaction with Clinical Laboratory Services at Public Hospitals in Ethiopia: A National Survey 

Dear Dr. Hailu:

I am pleased to inform you that your manuscript has been deemed suitable for publication in PLOS ONE. Congratulations! Your manuscript is now with our production department. 

With kind regards,

on behalf of

Prof Massimo Ciccozzi 

Academic Editor

PLOS ONE